# Recent Advances and Applications of Spiral Dynamics Optimization Algorithm: A Review

**Madiah Binti Omar** [1] , **Kishore Bingi** [2,*] , **B Rajanarayan Prusty** [2] and **Rosdiazli Ibrahim** [3]

1   Department of Chemical Engineering, Universiti Teknologi PETRONAS, Seri Iskandar 32610, Malaysia; madiah.omar@utp.edu.my
2   School of Electrical Engineering, Vellore Institute of Technology, Vellore 632014, India; b.r.prusty@ieee.org
3   Department of Electrical and Electronics Engineering, Universiti Teknologi PETRONAS, Seri Iskandar 32610, Malaysia; rosdiazli@utp.edu.my
*   Correspondence: kishore.bingi@vit.ac.in

**Abstract:** This paper comprehensively reviews the spiral dynamics optimization (SDO) algorithm and investigates its characteristics. SDO algorithm is one of the most straightforward physics-based optimization algorithms and is successfully applied in various broad fields. This paper describes the recent advances of the SDO algorithm, including its adaptive, improved, and hybrid approaches. The growth of the SDO algorithm and its application in various areas, theoretical analysis, and comparison with its preceding and other algorithms are also described in detail. A detailed description of different spiral paths, their characteristics, and the application of these spiral approaches in developing and improving other optimization algorithms are comprehensively presented. The review concludes the current works on the SDO algorithm, highlighting its shortcomings and suggesting possible future research perspectives.

**Keywords:** advances of SDO; applications of SDO; metaheuristic optimization; nature-inspired algorithms; optimization problems; spiral dynamics optimization; spiral-inspired optimization algorithms; spiral paths





## 1. Introduction

In engineering applications, metaheuristic optimization algorithms are more popular and widely used for computing the optimal solution [1]. This broad application is because:

1.   The algorithms are easy to implement and do not require gradient information as they depend on relatively simple concepts;
2.   The algorithms can avoid settling at optimal local solutions;
3.   The algorithms can be applied to various problems of different fields.

A great variety of nature and population-based metaheuristic optimization algorithms have been published in the literature [2]. As reported in [2], these algorithms are categorized into breeding-based, swarm intelligence-based, physics-based, chemistry-based, social human behavior-based, plant-based, and others. Many developed metaheuristic optimization algorithms published in the literature are swarm intelligence-based algorithms. After swarm intelligence-based algorithms, physics-based algorithms are the most widely proposed and implemented in various applications [3,4]. As the name suggests, in swarm intelligence-based algorithms, some degree of intelligence is present in the algorithm process while finding the optimal solution. However, in physics-based algorithms, the algorithm process is based on specific laws or principles [3,5,6]. The main advantage of physics-based algorithms compared to others is the most straightforwardness. This is because the algorithm's strategy is based on fundamental physical principles. Thus, the algorithms can consistently and accurately represent the dynamics over the entire domain. Further, some physics-based algorithms also take advantage of a nature-inspired

ratio, called the golden ratio, which helps to converge quickly and effectively when finding the optimal solution [7].

The most popular physics-based optimization algorithms are harmony search, gravitational search algorithm (GSA), big bang big crunch, electromagnetic field optimization (EFO), galaxy-based search [8], ray optimization, magnetic optimization, spiral dynamics optimization [9], and water cycle optimization [10]. Spiral dynamics optimization (SDO) is one of the most straightforward physics-based algorithms proposed by Tamura and Yasuda in 2011, developed using a logarithmic spiral phenomenon in nature [9]. The algorithm is simple and has few control parameters. Moreover, the algorithm has fast computational speed, local searching capability, diversification in the early phase, and intensification in the later stage.

This review paper provides the origin and concept of the SDO algorithm for an $n$-dimensional system. The effect of variation of spiral parameters (radius and angle) for two- and three-dimensional systems are analyzed by generating the conventional and hypotrochoid spiral trajectories. Besides, the recent advances in SDO algorithm, including adaptive, improved, and hybrid versions, are highlighted. The current applications of SDO and its variants are also focused. Different types of spirals, coordinates on $xy$-plane, and trajectories are generated to understand spiral behaviors. Further, various novel optimization algorithms' developments using these spirals are presented comprehensively. Therefore, this review paper helps in guiding multiple researchers who are currently working and willing to work by employing SDO and its variants to solve various engineering problems. Moreover, the review helps in developing or improving existing algorithms using the spiral phenomenon.

The paper's remaining sections are organized as follows: the origin and concept of the SDO algorithm and the effect of the spiral parameter in developing search trajectories are presented in Section 2. Section 3 offers the recent adaptive, improved, and hybrid versions of the SDO algorithm. Section 4 gives the different types of spiral trajectories and a list of novel optimization algorithms created using these trajectories. The applications of SDO and its hybrid versions are presented in Section 5. Finally, the paper is concluded in Section 6.

## 2. Spiral Dynamics Optimization Algorithm

This section presents the origin and the concept of the SDO algorithm for two-dimensional and three-dimensional systems. A detailed analysis of the effect of varying spiral parameters (radius and angle) is also presented.

### 2.1. Origin

Tamura and Yoshida developed the SDO algorithm in 2011 to mimic the spiral phenomena in nature [9,11]. Many spirals are available in nature, such as galaxies, aurora, blackbuck horns, hurricanes, tornadoes, seashells, snails, ammonites, cabbage butterflies, Pieris brassicae, chameleon tail, seahorse, and fish vortex [12,13]. The spirals are also seen in ancient art created by humanity during 5000 BC to 1600 AD [12]. Over the years, several researchers have made efforts to understand the spiral sequences and complexities and develop equations and algorithms of the spirals. Moreover, it is worth highlighting that the frequently encountered spiral phenomenon in nature is the logarithmic spiral, which can be seen in galaxies, tropical cyclones, and nautilus shells [14]. The discrete processes of generating a logarithmic spiral have been realized as an effective search behavior in metaheuristics, which inspired the spiral dynamics optimization algorithm to develop.

### 2.2. Concept

In the SDO algorithm, the multipoint search function for an $n$-dimensional system is formulated as [15],

$$x_{k+1} = rR^{(n)}(\theta)x_k - (rR^{(n)}(\theta) - I_n)x^*, \tag{1}$$

where $r$ is the spiral radius, $R^{(n)}(\theta)$ is the rotational matrix of order $n \times n$, $\theta$ is the spiral rotation angle, $I_n$ is the identity matrix of order $n \times n$, $x^*$ is the spiral center, $x_k$ and $x_{k+1}$ are the search point positions at iterations $k$ and $k+1$, respectively.

The rotational matrix $R^{(n)}(\theta)$ for an $n$-dimensional case on an arbitrary $x_i x_j$-plane is given as [9,16,17],

$$
R^{(n)}(\theta) = \begin{bmatrix}
1 & 0 & 0 & \dots & 0 & 0 & 0 \\
0 & 1 & 0 & \dots & 0 & 0 & 0 \\
0 & 0 & \cos(\theta_{i,j}) & \dots & -\sin(\theta_{i,j}) & 0 & 0 \\
\vdots & \vdots & \vdots & \ddots & \vdots & \vdots & \vdots \\
0 & 0 & \sin(\theta_{i,j}) & \dots & \cos(\theta_{i,j}) & 0 & 0 \\
0 & 0 & 0 & \dots & 0 & 1 & 0 \\
0 & 0 & 0 & \dots & 0 & 0 & 1
\end{bmatrix},
\tag{2}
$$

where $\theta_{i,j}$ is the spiral rotation angle around the origin on $x_i x_j$-plane.

From (2), the only one possibility of rotational matrix $R^{(2)}(\theta)$ for a two-dimensional system on $x_1 x_2$-plane is given as follows:

$$
R^{(2)}(\theta) = \begin{bmatrix}
\cos(\theta) & -\sin(\theta) \\
\sin(\theta) & \cos(\theta)
\end{bmatrix}.
\tag{3}
$$

On the other hand, the three possible combinations of rotational matrix $R^{(3)}(\theta)$ for a three-dimensional system on $x_1 x_2$, $x_2 x_3$, and $x_1 x_3$-planes are respectively given as follows:

$$
R_{1,2}^{(3)}(\theta) = \begin{bmatrix}
\cos(\theta_{1,2}) & -\sin(\theta_{1,2}) & 0 \\
\sin(\theta_{1,2}) & \cos(\theta_{1,2}) & 0 \\
0 & 0 & 1
\end{bmatrix},
\tag{4}
$$

$$
R_{2,3}^{(3)}(\theta) = \begin{bmatrix}
1 & 0 & 0 \\
0 & \cos(\theta_{2,3}) & -\sin(\theta_{2,3}) \\
0 & \sin(\theta_{2,3}) & \cos(\theta_{2,3})
\end{bmatrix}, \text{ and}
\tag{5}
$$

$$
R_{1,3}^{(3)}(\theta) = \begin{bmatrix}
\cos(\theta_{1,3}) & 0 & -\sin(\theta_{1,3}) \\
0 & 1 & 0 \\
\sin(\theta_{1,3}) & 0 & \cos(\theta_{1,3})
\end{bmatrix}.
\tag{6}
$$

From (1), it is to be noted that the model generated the spiral trajectories around the center $x^*$ and these trajectories are classified into two types [18,19]:

- If $r > 1$ and $\theta \in (-\frac{\pi}{2}, \frac{\pi}{2})$, the trajectory is a conventional spiral;
- If $r < 1$ and $\theta \in (-\frac{\pi}{2}, \frac{\pi}{2})$, the trajectory is a hypotrochoid spiral.

From the above classification, the spiral's direction of rotation based on the value of $\theta$ is classified as follows:

- If $\theta \in (-\frac{\pi}{2}, 0)$, the rotation of trajectory is clockwise;
- If $\theta \in (0, \frac{\pi}{2})$, the rotation of trajectory is anticlockwise.

The spiral trajectories for a two-dimensional system for various values of $r \in [-1, 1]$ and $\theta = \frac{\pi}{8}$ is shown in Figure 1. Similarly, the trajectories for various values of $\theta \in [-\frac{\pi}{2}, \frac{\pi}{2}]$ and $r = 0.85$ for conventional spiral and $r = -0.85$ for hypotrochoid spiral are shown in Figure 2. Further, the conventional and hypotrochoid spiral trajectories for both positive and negative values of $\theta$ are shown in Figure 3. In all these cases, the starting point used in the study is $(25, 25)$.

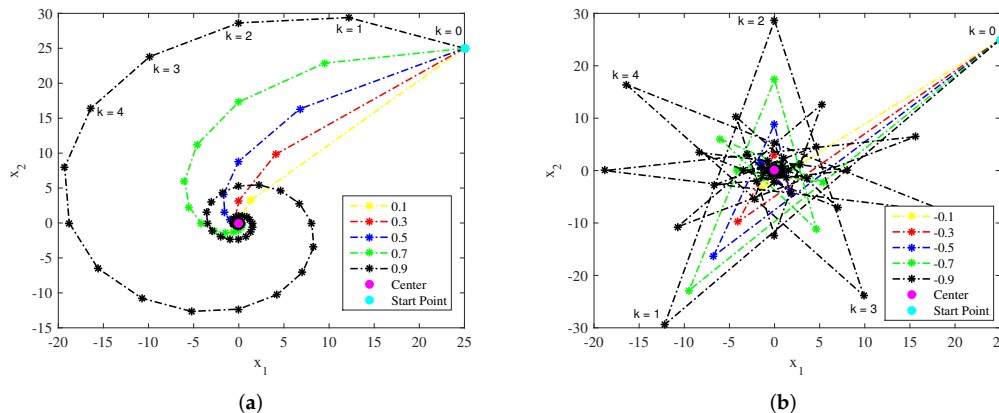

**Figure 1.** Spiral trajectories for a two-dimensional system for various values of $r \in [-1, 1]$ and $\theta = \frac{\pi}{8}$: (**a**) conventional spiral and (**b**) hypotrochoid spiral.

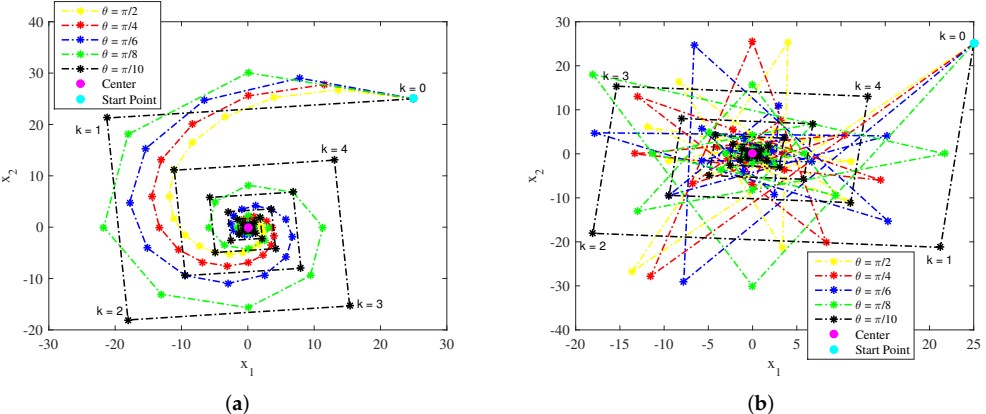

**Figure 2.** Spiral trajectories for a two-dimensional system for various values of $\theta \in \left[-\frac{\pi}{2}, \frac{\pi}{2}\right]$ and $r = 0.85$ for conventional spiral in (**a**) and $r = -0.85$ for hypotrochoid spiral in (**b**).

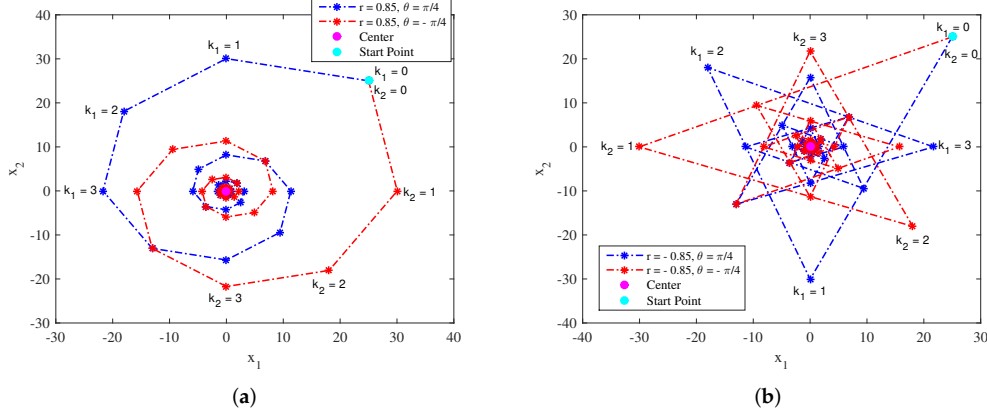

**Figure 3.** Spiral trajectories for a two-dimensional system for both positive and negative values of $\theta$: (**a**) conventional spiral and (**b**) hypotrochoid spiral.

Observing the notations $k = 0, k = 1, \ldots, k = 4$ on spiral trajectories in Figures 1–3, it can be noted that at each iteration, the spiral point from the starting point moves by an angle $\theta$ and then tends towards the center $x^*$. Thus, the net effect is the spiral movement of the initial point towards the center. The trajectories also depict the angle $\theta$, controlling the spiral curve. A smoother curve is achieved for smaller values of $\theta$, compared to the

boxy curved with larger values of $\theta$ (refer to Figure 2a). The spiral trajectories in Figure 3 show the clockwise and anticlockwise spiral movement for negative and positive angles, respectively. On the other hand, the spiral radius $r$ controls the spiral movement towards the center $x^*$. A quick movement of spiral towards the center is achieved for smaller values of $r$, compared to the slow movement with larger values of $r$ (refer to Figures 1 and 2). The hypotrochoid spirals shown in Figures 1b, 2b, and 3b are internal trajectories which are generated along a circle. The advantage of a hypotrochoid spiral over conventional spirals is it does not exceed the search space and can search most of the area in the search space.

In a similar way, the conventional and hypotrochoid spiral trajectories for a three-dimensional system with $r = 0.95$ and $\theta = \frac{\pi}{4}$ are shown in Figure 4. The trajectory in Figure 4a on the $x_1 x_2$-plane is obtained using the rotational matrix in (4). Similarly, the trajectories in Figure 4b,c on the $x_2 x_3$ and $x_1 x_3$-planes are obtained using the rotational matrices in (5) and (6), respectively. The starting point used is $(25, 25, 25)$ in all of these cases. The trajectories depict the conventional spiral with a positive $r$ value and the hypotrochoid spiral with a negative $r$ value. As the $\theta$ value is positive, all the spiral movements are anticlockwise. As mentioned earlier, the advantage of hypotrochoid spirals is they can search most of the area in the search space, as shown in Figure 4. The search space of a conventional spiral is only on the positive plane, while the hypotrochoid spirals search space is both negative and positive. Thus, the trajectories in the figure conclude that the hypotrochoid spirals can search most of the area in the search space.

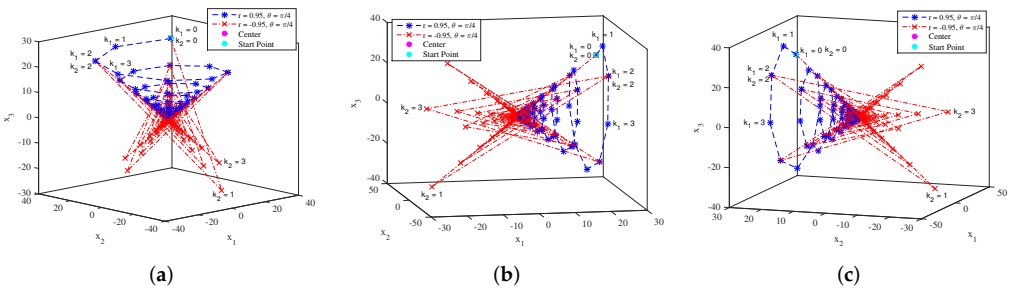

(a)           (b)           (c)

**Figure 4.** Conventional and hypotrochoid spiral trajectories for a three-dimensional system with $r = 0.95$ and $\theta = \frac{\pi}{4}$: (**a**) on $x_1 x_2$-plane with $R_{1,2}$. (**b**) on $x_2 x_3$-plane with $R_{2,3}$. (**c**) on $x_1 x_3$-plane with $R_{1,3}$.

## 3. Advances of Spiral Dynamics Optimization Algorithm

This section presents the recent adaptive, improved, and hybrid versions of the SDO algorithm.

### 3.1. Adaptive Versions of Spiral Dynamics Optimization Algorithm

Researchers have developed the adaptive versions of the SDO algorithm by dynamically varying the spirals' radius and angle based on the fitness value during each iteration. The four types of proposed adaptive approaches in the literature are linear, quadratic, exponential, and fuzzy [16,20,21]. The mathematical functions of spirals' radius and angle using the proposed approaches are given in Figure 5.

In the figure, the notations are defined as follows:

- $r_{la}$ and $\theta_{la}$ are the computed radius and angle using linear adaptive approach;
- $r_{qa}$ and $\theta_{qa}$ are the obtained radius and angle using quadratic adaptive approach;
- $r_{ea}$ and $\theta_{ea}$ are the radius and angle obtained using exponential adaptive approach;
- $r_{fa}$ and $\theta_{fa}$ are the calculated radius and angle using fuzzy adaptive approach;
- $r_l \in [0, 1]$ and $r_u \in [0, 1]$ are the minimum and maximum radius of spiral;
- $\theta_l \in [0, 1]$ and $\theta_u \in [0, 1]$ are the minimum and maximum angles of spiral;
- $c_1$ and $c_2$ are constants;
- fuzzy$(\cdot)$ is the fuzzy logic mapping;

- $Y_{\text{Fit}}$ is the difference between fitness value at a current iteration $f(x_i(k))$ and best fitness $\min(f(x_i(k)))$, is defined as,

$$Y_{\text{Fit}} = f(x_i(k)) - \min(f(x_i(k))). \tag{7}$$

In [17], using the linear adaptive approach in Figure 5, the authors have proposed the adaptive hypotrochoid SDO algorithm. The proposed algorithm performs best on various benchmark functions compared to conventional techniques. On the other hand, in [22], a self-adaptive approach is proposed for the SDO algorithm to update the spiral radius and angle during the optimization. The approach's advantage is that all search points are updated by randomly tuning the parameter values in each iteration. Similarly, the authors of [23] have proposed an adaptive SDO by incorporating three mechanisms, such as (i) bi-considering updation, (ii) self-adaptive radius, and (iii) punish mechanisms. The proposed algorithm boosted the optimization efficiency and avoided trapping at the local optimal minima.

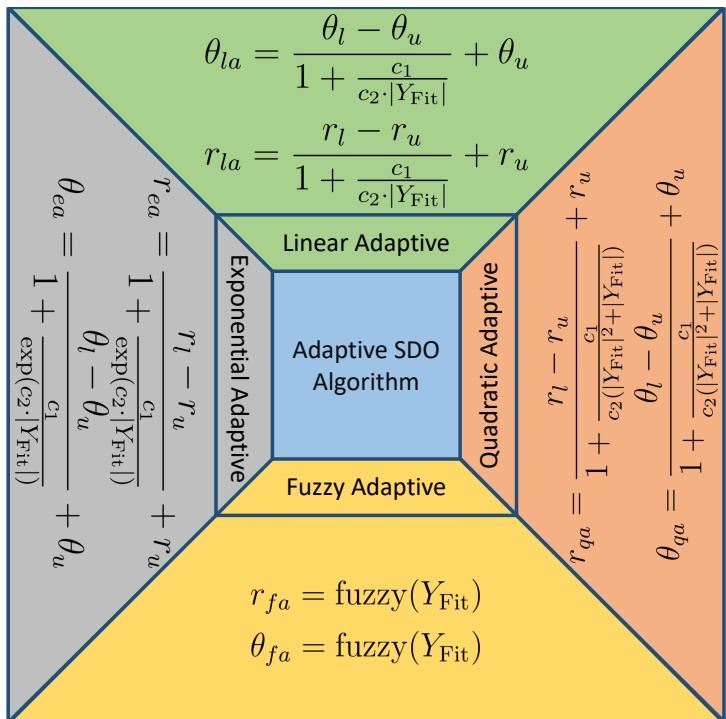

**Figure 5.** Adaptive versions of the SDO algorithm.

### 3.2. Improved Versions of Spiral Dynamics Optimization Algorithm

As mentioned earlier in Section 2.2, the algorithm settles into optimal local values at the end of the optimization process due to insufficient exploration of the conventional SDO's search space. Thus, to avoid this problem, Nasir et al. have proposed the improved SDO algorithm using the bacterial foraging algorithms' elimination–dispersal strategy [24,25]. In this enhanced version, the algorithm structure is kept the same. However, two new phases, namely elimination and dispersal, are introduced. Similarly, Hashim et al. have proposed the chaotic SDO algorithm logistic chaotic map patterns in the conventional SDO [26,27]. The chaotic map pattern helps in the initial population distribution rather than randomly in conventional SDO. Moreover, the search strategy of the artificial bee colony optimization algorithm is employed to improve the SDO's exploration capability. The authors have also proposed the greedy SDO algorithm by incorporating the greedy selection stage and chaotic logistic map in the conventional SDO [28]. In this selection stage, the obtained solution is compared to the previous value for updating the spiral positions. The authors of [18,19] have proposed the hypotrochoid SDO algorithm in which the search points follow the hypotrochoid spiral rather than the conventional spiral in SDO.

The proposed hypotrochoid SDO can explore the search space more effectively and explore the whole neighborhood of the optimal center. The experimental validation on optimal triaxial accelerometers placement in the Shanghai Tower in China [19], and sizing and layout of truss structures [18] has shown the better performance of hypotrochoid SDO than its predecessors.

The SDO algorithm in Section 2.2 is developed by utilizing a feature of the logarithmic spiral. This algorithm is also known as a deterministic or direct-solving metaheuristic optimization algorithm. One of the significant drawbacks of this algorithm is the slow convergence. Therefore, the authors of [29–31] have proposed a stochastic SDO algorithm by incorporating some random disturbances at each searching point of the algorithm. Similarly, the authors of [32] have introduced the iterative SDO algorithm for analyzing the information on blurred images. In this algorithm, the model's output is given as an input to the same model iteratively. Thus, the optimization algorithm searches for the sharp image spirally with the blurred vision at the initial stage. On the other hand, the authors of [33] have proposed the distributed SDO algorithm to increase the diversity in the search space. In this conventional SDO algorithm, it is clear that the search points rotate spirally around the optimal center only. Thus, the algorithm falls into the local minimum quickly. However, in the proposed distributed SDO algorithm, the population of search points is split into sub-populations to increase diversity and capture the whole search space. The summary of all these approaches is given in Figure 6.

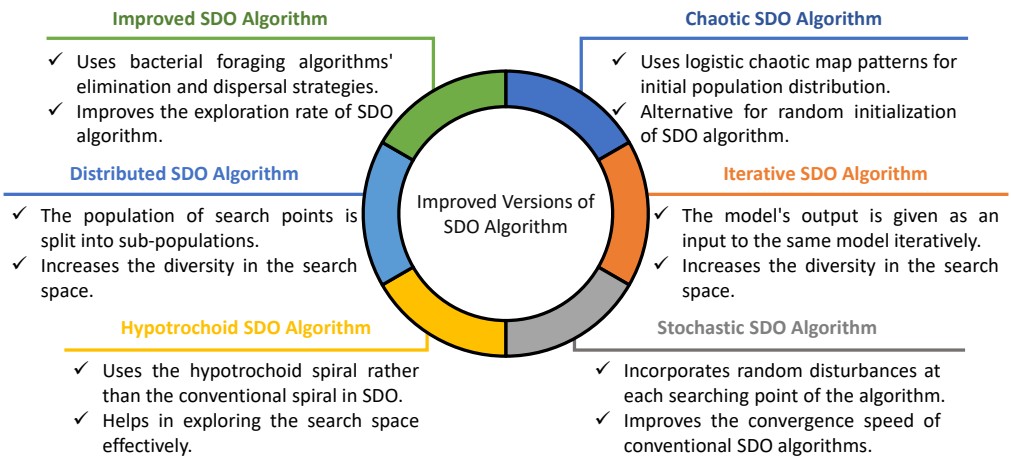

**Figure 6.** Improved versions of the SDO algorithm.

### 3.3. Hybrid Versions of Spiral Dynamics Optimization Algorithm

From the literature review, the following points are worth highlighting on the performance of the SDO algorithm. SDO has the advantages of a simple structure, few control parameters, and early diversification and intensification strategies. However, the SDO's performance is poor in searching the whole search space [20,34], and the exploration mechanism of the SDO needs to be improved [35]. The algorithm gets trapped at optimal local minima easily [33].

Thus, to improve the performance of SDO, researchers have proposed the hybridization of SDO with other algorithms. Further, various algorithms' performance has also been enhanced using SDO. The hybrid versions of the SDO algorithm presented in the literature used an artificial bee colony (ABC) [36,37], antlion optimization (ALO) [38], bacterial chemotaxis algorithm (BCA) [20,34,39], bacterial foraging algorithm (BFA) [35,40,41], biogeography-based optimization (BBO) [42], cuckoo search (CS) [43], genetic algorithm (GA) [44], particle swarm optimization (PSO) [45–48], sine-cosine algorithm (SCA) [49], and teaching–learning-based optimization (TLBO) [50], as shown in Figure 7. As shown in the figure, the excellent exploitation strategy of SDO is hybridized with the fast exploration strategy of another algorithm to balance both the exploitation and exploration phases.

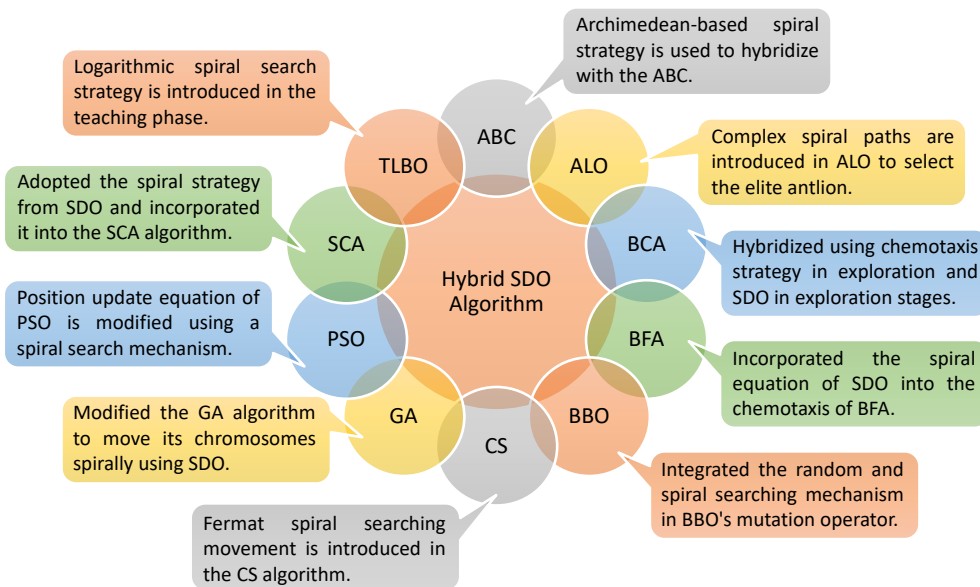

**Figure 7.** Hybrid versions of the SDO algorithm.

Moreover, there are several other novel optimization algorithms in which spiral behavior or trajectory is used during the development of the algorithm. A detailed description of various spiral paths and a list of novel spiral path-inspired optimization algorithms are discussed in the following section.

## 4. Spiral Path Inspired Optimization Algorithms

The first part of this section presents the various spiral trajectories used to develop the optimization algorithms. Then, the list of different novel optimization algorithms created using these spirals is shown.

### 4.1. Spiral Paths

Patterns referred to as visible consistencies found in nature are trees, spirals, waves, etc. Visual patterns in nature are modeled using chaos theory, fractals, spirals, etc. In some natural patterns, the spirals and fractals are related. For instance, a variant of the logarithmic spiral, namely the Fibonacci spiral, is based on the golden ratio and Fibonacci numbers. As it is logarithmic, the curve at every scale appears the same and can be considered a fractal. Romanesco broccoli is an example of such Fractal spirals. The above patterns inspired researchers to develop optimization algorithms. Different types of spiral trajectories used in the research include:

- Archimedes spira;
- Cycloid spiral;
- Epitrochoid spiral;
- Hypotrochoid spiral;
- Logarithmic spiral;
- Rose spiral;
- Inverse spiral; and
- Overshoot spirals.

A detailed description of the five most widely used spirals, including Archimedes, logarithmic, rose, epitrochoid, and hypotrochoid, is provided underneath. This detailed description includes the coordinates on the $xy$-plane and trajectories showing the effect of each parameter on the $xy$-plane.

### 4.1.1. Logarithmic Spiral

The logarithmic spirals often appear in nature. For instance, the nautilus cutaway, Iceland's low-pressure area, galaxies, and tropical cyclones arms usually take a logarithmic spiral shape. The logarithmic spiral is also known as equiangular or growth spiral because the spiral distance increases in geometric progression. The coordinates of a logarithmic spiral on *xy*-plane are given as follows [13,38]:

$$x(\phi) = a \cdot e^{b\phi} \cdot \cos(\phi), \ y(\phi) = a \cdot e^{b\phi} \cdot \sin(\phi), \tag{8}$$

where $\phi$ is the angle, $a$ and $b$ are the arbitrary constants.

The logarithmic spiral for $a = 0.18$, $\phi$ from $-4\pi$ to $4\pi$, and various $b$ values is shown in Figure 8. The spiral in Figure 8a is obtained for positive values of $b$, while Figure 8b is obtained for negative values. The trajectories in Figure 8 show that parameter $b$ controls the tightness and the direction of the spiral. The trajectories in Figure 8a also depict the logarithmic spiral proprieties that for positive $b$ values and $\phi$ tends to $+\infty$, the spiral evolves in an anticlockwise direction. Whereas for the same $b$ values and $\phi$ tends to $-\infty$, the spiral evolves in a clockwise direction. However, for negative $b$ values, the spiral evolves or twists in the opposite direction.

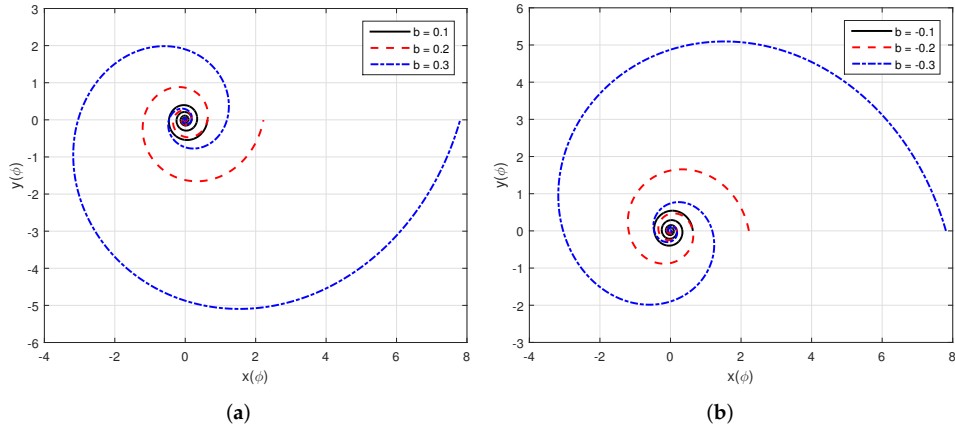

(a) (b)

**Figure 8.** Logarithmic spiral with various values of $b$: (**a**) logarithmic spiral with positive $b$ values and (**b**) logarithmic spiral with negative $b$ values.

### 4.1.2. Archimedean Spiral

Archimedean spiral is another famous spiral that has been used in significant applications of engineering, biology, etc. The Archimedean spiral is also known as the arithmetic spiral. This spiral can be seen in nature in ferns, millipedes, and human fingerprints. The spiral trajectory is the locus of a point's position that moves away from the fixed point with a constant speed along a line that rotates with a constant angular velocity. The coordinates of an Archimedean spiral on *xy*-plane is given as follows [13,38]:

$$x(\psi) = (c + d \cdot \psi) \cdot \cos(\psi), \ y(\psi) = (c + d \cdot \psi) \cdot \sin(\psi), \tag{9}$$

where $c$ and $d$ are constants that define the spirals initial radius and the successive turns difference, respectively.

The Archimedean spiral for $c = 0.5$, $\psi$ from 0 to $-7\pi$, and various $d$ values are shown in Figure 9. The trajectory in Figure 9a is obtained for positive values of $d$, while Figure 9b is obtained for negative values. As the initial radius is $c = 0.5$, all the spirals are starting at this value, as shown in Figure 9. The spiral growth rate $d$ controls the increment per revolution. Thus, the distance between successive turns is constant, which is equal to the value of $d$. Moreover, the parameter $d$ controls the evolution of the spiral. The spiral in

Figure 9a depicts that for positive *d* values, and the spiral evolves in an anticlockwise direction. Whereas for negative *d* values, the spiral evolves clockwise.

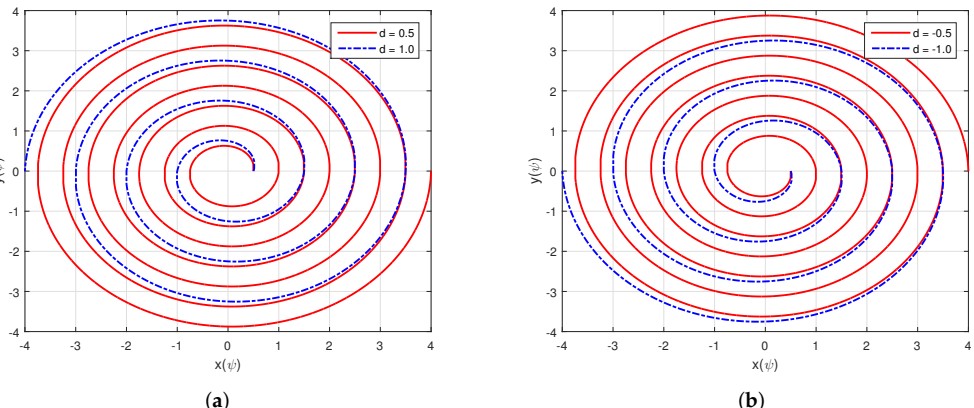

**Figure 9.** Archimedean spiral with various values of *d*: (**a**) Archimedean spiral with positive *d* values and (**b**) Archimedean spiral with negative *d* values.

Observing the spirals in Figures 8 and 9 shows a difference between the Archimedean and logarithmic spirals worth highlighting. In the Archimedean spiral, the intersection points of a ray from the origin on successive turnings have a constant separation distance. However, in a logarithmic spiral, these distance of intersection points on next turnings from the origin will form a geometric progression.

### 4.1.3. Rose Spiral

As the name suggests, the rose spiral is often seen in the unfurling of rose petals and holds the properties of symmetric and periodic arc curves. The coordinates of a rose spiral on *xy*-plane is given as follows [13,38]:

$$x(\xi) = e \cdot \cos(n\xi) \cdot \cos(\xi), \quad y(\xi) = e \cdot \cos(n\xi) \cdot \sin(\xi), \tag{10}$$

where *e* and *n* are constants that define the pedal length and number, respectively.

The rose spiral with various values of *e* and *n* are shown in Figure 10. The spiral in Figure 10a is achieved for *n* = 2 and multiple values of *e*. Similarly, the spiral in Figure 10b is obtained for *e* = 2 and various values of *n*. In both cases, *ξ* ranges from 0 to 2. The spirals in Figure 10a depict that parameter *e* controls the petal length. It is worth noting that as the value of *e* increases, the petal length increases. The spirals in Figure 10b also show that *n* controls petals' number, size, and length. For an even value of *n*, the number of petals is 2*n*. However, for odd values of *n*, the number of petals is only *n*.

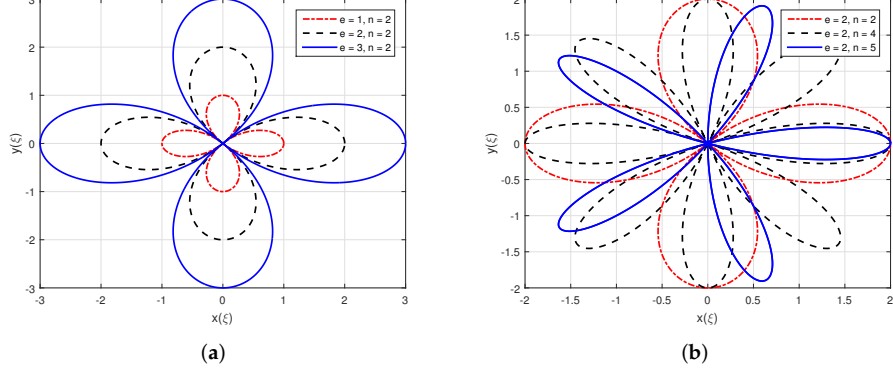

**Figure 10.** Rose spiral with various values of *e* and *n*: (**a**) rose spiral with constant *n* value and variable *e* and (**b**) rose spiral with constant *e* value and variable *n*.

### 4.1.4. Epitrochoid and Hypotrochoid Spirals

Epitrochoid and hypotrochoid spirals are a family of curves generated by a point attached to a rolling circle. This rolling circle will roll out around the outside of a fixed circle to form an epitrochoid spiral. On the other hand, to create a hypotrochoid spiral, the rolling one will roll around inside the fixed one. Let $\rho_1$ and $\rho_2$ be the radii of rolling and fixed circles, respectively, and $f$ is the distance between the point and rolling circle's center. The coordinates of epitrochoid spiral on $xy$-plane is given as [13,38],

$$x(\zeta) = (\rho_2 + \rho_1) \cdot \cos(\zeta) - f \cdot \cos\left(\frac{\rho_2 + \rho_1}{\rho_1}\zeta\right), \text{ and}$$

$$y(\zeta) = (\rho_1 + \rho_2) \cdot \sin(\zeta) - f \cdot \sin\left(\frac{\rho_1 + \rho_2}{\rho_1}\zeta\right).$$

(11)

Similarly, the coordinates of a hypotrochoid spiral on $xy$-plane is given as follows:

$$x(\zeta) = (\rho_2 - \rho_1) \cdot \cos(\zeta) + f \cdot \cos\left(\frac{\rho_2 - \rho_1}{\rho_1}\zeta\right), \text{ and}$$

$$y(\zeta) = (\rho_2 - \rho_1) \cdot \sin(\zeta) - f \cdot \sin\left(\frac{\rho_2 - \rho_1}{\rho_1}\zeta\right).$$

(12)

The trajectories of epitrochoid and hypotrochoid spirals for $\rho_1 = 0.8$, $\rho_2 = 3$, $d = 2.5$, and $\zeta$ ranging from 0 to $10\pi$ is shown in Figure 11a,b, respectively. In both spirals, it should be noted that $\zeta$ significantly affects the spiral's shape. If the considered $\zeta$ ranges from 0 to $2\pi$, the rolling circle will revolve only once around the fixed circle. Thus, it is not possible to obtain the whole pattern of the spiral. These spirals can be drawn using Spirograph toys and often appear in nature. For instance, the planets orbit in a geocentric system, and Wankel engines' combustion chambers take these spiral shapes.

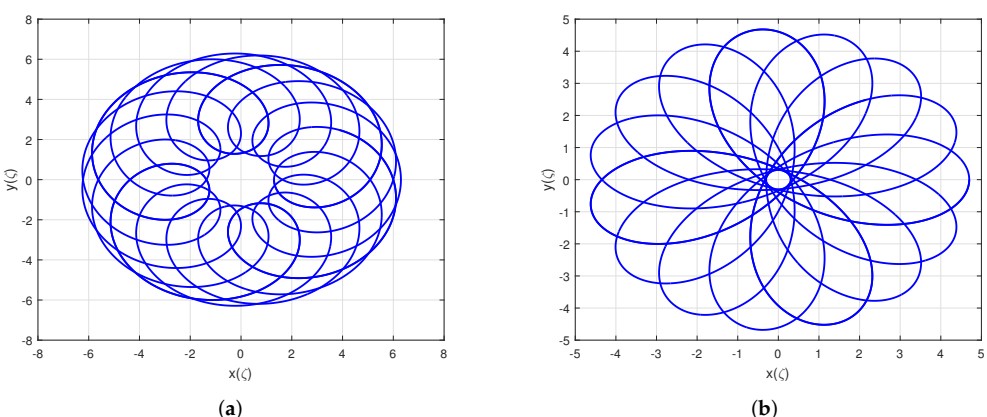

(a)                                                          (b)

**Figure 11.** Epitrochoid and hypotrochoid spirals for $\rho_1 = 0.8$, $\rho_2 = 3$, and $d = 2.5$: (**a**) epitrochoid spiral and (**b**) hypotrochoid spiral.

### 4.2. Spiral Path-Based Optimization Algorithms

Over the years, researchers have developed various novel optimization algorithms in which the spiral motion has been used while mimicking the system's behavior. Further, an improved version of multiple algorithms is also proposed using spiral trajectories to improve the performance of conventional techniques. Table 1 provides the list of spiral path-inspired optimization techniques, including the inspiration of developing the algorithm, the type of spiral used, and the source code links.

**Table 1.** List of spiral path-inspired optimization algorithms.

| Ref. | Year | Algorithm | Author | Inspiration | Spiral Type | Source Code Link |
|------|------|-----------|--------|-------------|-------------|------------------|
| [40,41] | 2010 | Spiral Bacterial Foraging Optimization Algorithm | Alireza Kasaiezadeh et al. | *E. coli* bacteria foraging behavior | Spiral | – |
| [8] | 2011 | Galaxy-Based Search Algorithm | Hamed Shah-Hosseini | Arms of the spiral galaxy | Spiral | – |
| [51] | 2014 | Hurricane-Based Optimization Algorithm | Isamil Rbouh et al. | Behavior of hurricanes, radial wind, and pressure profiles | Logarithmic spiral | – |
| [52] | 2015 | Moth–flame Optimization Algorithm | Seyedali Mirjalili | Moths' navigation behavior around the flame | Logarithmic spiral | https://seyedalimirjalili.com/mfo (accessed on 1 December 2021) |
| [53] | 2016 | Whale Optimization Algorithm | Seyedali Mirjalili | Whales' hunting bubble net phenomenon | Logarithmic spiral | https://seyedalimirjalili.com/woa (accessed on 1 December 2021) |
| [54] | 2017 | Moth Swarm Optimization Algorithm | Al-Attar Ali Mohamed et al. | Moths orientation towards the moonlight | Logarithmic spiral | https://mathworks.com/matlabcentral/fileexchange/57822 (accessed on 1 December 2021) |
| [50] | 2018 | Improved Teaching–Learning-Based Optimization Algorithm | Zhuoran Zhang et al. | effect of teacher influence on learners | Logarithmic spiral | – |
| [55] | 2018 | Developed Grey Wolf Optimization Algorithm | Mostafa Abdo et al. | Grey wolves leadership and hunting strategies | Logarithmic spiral | – |
| [56] | 2019 | Seagull Optimization Algorithm | Gaurav Dhiman et al. | Seagulls' migration and attacking behavior | 3D logarithmic spiral | https://mathworks.com/matlabcentral/fileexchange/75180 (accessed on 1 December 2021) |
| [57] | 2019 | Emperor Penguins Colony Optimization Algorithm | Sasan Harifi et al. | Emperor penguins behavior | Logarithmic spiral | – |
| [36,37] | 2019 | Spiral Artificial Bee Colony Algorithm | Sonal Sharma et al. | Honey bee swarms' intelligent foraging behavior | Logarithmic and Archimedean spirals | – |
| [58] | 2019 | Sooty Tern Optimization Algorithm | Gaurav Dhiman et al. | Sooty terns' migration and attacking behaviors | Spiral | https://mathworks.com/matlabcentral/fileexchange/76667 (accessed on 1 December 2021) |
| [59] | 2019 | Whirlpool Algorithm | Yuanyang Zou et al. | Physical phenomenon of whirlpool | Spiral | – |
| [60] | 2020 | Improved Crow Search Algorithm | Xiaoxia Han et al. | Crows intelligent behavior of searching, hiding and retrieving food | Logarithmic Spiral | – |

**Table 1.** *Cont.*

| Ref. | Year | Algorithm | Author | Inspiration | Spiral Type | Source Code Link |
|---|---|---|---|---|---|---|
| [38] | 2020 | Improved Ant Lion Optimization Algorithm | M. W. Guo et al. | Natural hunting phenomenon of antlions | Archimedes, Cycloid, Epitrochoid, Hypotrochoid, Logarithmic, Rose, Inverse, and Overshoot spirals | – |
| [61] | 2020 | Manta Ray Foraging Optimization Algorithm | Weiguo Zhao et al. | Manta rays intelligent behavior | Logarithmic spiral | https://www.mathworks.com/ matlabcentral/fileexchange/73130 (accessed on 1 December 2021) |
| [62] | 2020 | Bald Eagle Search Optimization Algorithm | H. A. Alsattar et al. | Bald eagles hunting behavior | Spiral | https://mathworks.com/ matlabcentral/fileexchange/86862 (accessed on 1 December 2021) |
| [63] | 2020 | Improved Firefly Algorithm | Jinran Wu et al. | Fireflies' flashing behavior | Logarithmic spiral | https://github.com/wujrtudou/ AdaptiveFireflyAlgorithm (accessed on 1 December 2021) |
| [64] | 2021 | Spiral Water Cycle Algorithm | Heba F. Eid et al. | Natural hydrological cycle process | Hyperbolic spiral | – |
| [65] | 2021 | Improved Slap Swarm Optimization Algorithm | Diab Mokeddem | Behavior of slap chains | Logarithmic spiral | – |
| [66] | 2021 | Spiral Flying Sparrow Search Algorithm | Chengtian Ouyang et al. | Sparrow's behaviors during group wisdom, antipredation, and foraging | Logarithmic spiral | – |
| [67] | 2021 | Spiral Grasshopper Optimization Algorithm | Zhangze Xu et al. | Grasshoppers foraging and swarming behavior | Logarithmic spiral | – |
| [68] | 2021 | Aquila Optimization Algorithm | Laith Abualigah et al. | Aquilas' behavior during prey catching | Spiral | https://www.mathworks.com/ matlabcentral/fileexchange/89381 (accessed on 1 December 2021) |
| [69] | 2021 | Spiral Spotted Hyena Optimization Algorithm | Vijay Kumar et al. | Spotted hyenas behavior during hunting | Logarithmic spiral | – |
| [70] | 2021 | Spiral Chicken Swarm Optimization Algorithm | Miao Li et al. | Chicken swarms' hierarchical order and its behaviors | Logarithmic spiral | – |
| [71] | 2021 | Golden Eagle Optimization Algorithm | Abdolkarim Mohammadi- Balani et al. | Golden eagles' intelligent behavior during hunting | Spiral | https://mathworks.com/ matlabcentral/fileexchange/84430 (accessed on 1 December 2021) |

For example, a detailed description of four novel optimization algorithms in which spiral trajectory has been used in the development is explained underneath. The chosen novel optimizations algorithms list includes moth–flame, whale, seagull, and Aquila. Further, a detailed description of four improved optimization algorithms using spiral trajectories is also explained in this section. The enhanced optimization algorithms are the water cycle, antlion, slap swarm, and sparrow search. Some of these algorithms have been widely used by various researchers recently, and others have been developed newly, thus selected for the detailed explanation.

### 4.2.1. Moth–Flame Optimization Algorithm

The moth–flame optimization algorithm was developed in 2015 by Seyedali Mirjalili from the behavior of moths' navigation around the light/flame in a spiral path [52,72,73]. The application of a logarithmic spiral to mimic the moths' transverse orientation property around the flame in this algorithm is explained underneath. In the algorithm, the initial moths' positions will be updated with respect to flames using the logarithmic spiral as follows [52,74]:

$$m_{i,j} = \begin{cases} D_{i,j} \cdot e^{b\tau} \cdot \cos(2\pi\tau) + f_{i,j}, & \text{for } i \leq F_N \\ D_{i,j} \cdot e^{b\tau} \cdot \cos(2\pi\tau) + f_{N,j}, & \text{for } i > F_N \end{cases}, \tag{13}$$

where $m_{i,j}$, $f_{i,j}$, and $D_{i,j}$ are the positions of $j$th variable of $i$th moth, flame, and distance between the moth and its corresponding flame, $N$ is the total number of flames. Further, $b$ and $\tau$ are the parameters of logarithmic spiral (refer to Section 4.1.1).

The major drawback of this algorithm is the premature convergence at optimal local solutions during the search process. Moreover, they cannot be applied to permutation problems as it is developed for continuous search space [75]. As mentioned in Table 1, the source code of this optimization algorithm created using MATLAB for both single and multiobjective problems is made publicly available by the developer on his website at https://seyedalimirjalili.com/mfo (accessed on 1 December 2021). Further, the links for the source code using other platforms, such as Python, C++, and R studio, are also available on the same website.

### 4.2.2. Whale Optimization Algorithm

The whale optimization algorithm is a novel metaheuristic algorithm developed in 2016 by Seyedali Mirjalili and Andrew Lewis to mimic whales' hunting bubble net phenomenon in a spiral motion [53,76–79]. The algorithm is a model of capturing whales' behavior during the encircling, attacking, and searching of prey. During the encircling phase, all the whales' positions will be updated to move towards the best whale position, which is near to the target and is given as,

$$\vec{X}(i+1) = \vec{X}^*(i) - \vec{A} \cdot |\vec{C} \cdot \vec{X}^*(i) - \vec{X}(i)|. \tag{14}$$

During the phase of attacking the prey, the whales move spirally using the bubble net movement phenomenon. Thus, position updation of whales during this phenomenon in logarithmic spiral motion is as follows:

$$\vec{X}(i+1) = |\vec{X}^*(i) - \vec{X}(i)| \cdot e^{bl} \cdot \cos(2\pi l)\vec{X}^*(i). \tag{15}$$

Finally, the whales will choose either encircling or attacking during the searching of prey, which can be achieved using the following model:

$$\vec{X}(i+1) = \begin{cases} \vec{X}^*(i) - \vec{A} \cdot |\vec{C} \cdot \vec{X}^*(i) - \vec{X}(i)|, & p < 0.5, \\ |\vec{X}^*(i) - \vec{X}(i)| \cdot e^{bl} \cdot \cos(2\pi l)\vec{X}^*(i), & p \geq 0.5. \end{cases} \tag{16}$$

Therefore, the position updation of all the whales during all three phases is summarized as,

$$\vec{X}(i+1) = \begin{cases} \begin{cases} \vec{X}^*(i) - \vec{A} \cdot |\vec{C} \cdot \vec{X}^*(i) - \vec{X}(i)|, & \vec{A} < 1, \\ \vec{X}_r(i) - \vec{A} \cdot |\vec{C} \cdot \vec{X}_r(i) - \vec{X}(i)|, & \vec{A} \geq 1, \end{cases} & p < 0.5, \\ |\vec{X}^*(i) - \vec{X}(i)| \cdot e^{bl} \cdot \cos(2\pi l)\vec{X}^*(i), & p \geq 0.5, \end{cases} \tag{17}$$

where the vectors $\vec{X}^*(i)$ is the closest whale's position to the prey, $\vec{X}(i)$ and $\vec{X}(i+1)$ are the whales' positions at $i$th and $i+1^{\text{th}}$ iterations, $\vec{A}$ and $\vec{C}$ are the coefficients, $b$ and $l$ are the parameters of logarithmic spiral (refer to Section 4.1.1). Further, it is to be noted that for $\vec{A} \geq 1$, positions updation has been achieved using $\vec{X}_r(i)$, a random position vector at $i$th iteration.

This whale optimization algorithm has the drawbacks of lower accuracy, slow convergence, and being trapped into optimal local solutions and cannot solve higher-dimensional problems effectively [80]. As given in Table 1, the source codes of this optimization algorithm for single-objective problems using MATLAB, Python, C++, and R are publicly available at https://seyedalimirjalili.com/woa (accessed on 1 December 2021).

### 4.2.3. Seagull Optimization Algorithm

Gaurav Dhiman et al. proposed the seagull optimization algorithm in 2019 to mimic the seagulls' migration and hunting behavior [56]. The algorithm is a mathematical model of seagulls' behavior in two stages, namely migration and attack. During the stage of natural attacking, the seagulls maintain spiral behavior in the air. The coordinates of this spiral behavior in $x$, $y$, and $z$ planes are modeled as follows:

$$x = u \cdot e^{kv} \cdot \cos(k), \; y = u \cdot e^{kv} \cdot \sin(k), \; z = u \cdot e^{kv} \cdot k, \tag{18}$$

where $k \in [0, 2\pi]$ is the spiral angle, $u$ and $v$ are the arbitrary constants.

The seagull optimization algorithm has the significant drawback of weak population diversity during the search process [81]. The link to the MATLAB-based source code of this optimization algorithm is given in Table 1.

### 4.2.4. Aquila Optimization Algorithm

The Aquila optimization algorithm was proposed in 2021 by Laith Abualigah et al. to mimic Aquila's behavior during prey catching [68]. The algorithm constitutes four stages: (i) expanded exploration, (ii) narrowed exploration, (iii) expanded exploitation, and (iv) narrowed exploitation. During the stage of narrowed exploration, the Aquila rotates over a target prey for a short glide attack. This behavior is modeled as follows:

$$X(t+1) = X_{best}(t) \cdot Levy() + X_r(t) + (y - x) \cdot rand(), \tag{19}$$

where $X_r(t)$ and $X_{best}(t)$ are the random and best solutions at $t$th iteration, $X(t+1)$ solution at $(t+1)^{\text{th}}$ iteration, $rand() \in (0, 1]$ is the random number, and $Levy()$ is the Lévy distribution. Further, $x$ and $y$ are the Cartesian coordinates of the spiral with radius $r$ and angle $l$ given as follows:

$$x = r \sin(l), \; y = r \cos(l). \tag{20}$$

From the above, it is to be highlighted that the Levy flight's effect is relatively weak. Thus, the algorithm has insufficient local exploitation ability [82]. The MATLAB and Java-based source code link of this optimization algorithm for single-objective problems is given in Table 1.

4.2.5. Water Cycle Optimization Algorithm

The water cycle optimization algorithm was proposed in 2012 by Eskandar et al. to mimic the natural hydrological cycle process [10,83,84]. The algorithm simulates the stream and river flow, rainfall, and evaporation into the sea. In this algorithm, the position update of (a) streams flow to the rivers, (b) streams flow to the sea, and (c) rivers flow to the sea are respectively given as follows:

$$X_{st}(i+1) = X_{st}(i) + rand() \cdot C \cdot (X_r(i) - X_{st}(i)), \tag{21}$$

$$X_{st}(i+1) = X_{st}(i) + rand() \cdot C \cdot (X_{se}(i) - X_{st}(i)), \tag{22}$$

$$X_r(i+1) = X_r(i) + rand() \cdot C \cdot (X_{se}(i) - X_r(i)), \tag{23}$$

where $X_{st}(i)$, $X_r(i)$ and $X_{se}(i)$ are the positions of stream, river, and sea at $i$th iteration, $X_{st}(i+1)$, $X_r(i+1)$, and $X_{se}(i+1)$ are the positions of stream, river, and sea at $(i+1)^{\text{th}}$ iteration, $C \in [1,2]$ is the constant value and $rand() \in (0,1]$ is the random number.

The MATLAB-based source code of this conventional optimization algorithm for both constrained and unconstrained problems, including several improved versions and multiobjective problems, are made publicly available by the researcher on his website at https://ali-sadollah.com/water-cycle-algorithm-wca/ (accessed on 1 December 2021).

The algorithm has insufficient exploitation ability, and thus, in [64], the authors have integrated the hyperbolic spiral, which helps improve the exploitation ability of the algorithm. Therefore, modified position update equations using the hyperbolic spiral are given as follows:

$$X_{st}(i+1) = X_{st}(i) + |X_r(i) - X_{st}(i)| \cdot \cos(2\pi l)/l, \tag{24}$$

$$X_{st}(i+1) = X_{st}(i) + |X_{se}(i) - X_{st}(i)| \cdot \cos(2\pi l)/l, \tag{25}$$

$$X_r(i+1) = X_r(i) + |X_{se}(i) - X_r(i)| \cdot \cos(2\pi l)/l, \tag{26}$$

where $l \in [-1,1]$ is the parameter of hyperbolic spiral, which is an uniformly distributed random number.

4.2.6. Ant Lion Optimization Algorithm

Seyedali Mirjalili proposed the antlion optimization algorithm in 2015 to mimic the natural hunting phenomenon of antlions [85–88]. The algorithm is a model of capturing the following ants and antlions behaviors: (i) the ants' random walk behavior and gets trapped in antlions pits and (ii) the antlions' hunting behaviors include building traps, sliding ants towards them, catching, rebuilding pits, and elitism. The algorithm retains the best antlion with optimal fitness value, elitism, and the corresponding antlion is called elite antlion. Thus, the elite and selected antlions update their position randomly as follows:

$$\text{Ant}_i(t) = \frac{R_e(t) + R_a(t)}{2}, \tag{27}$$

where $R_e(t)$ and $R_a(t)$ are the elite and selected antlions random walk during $t$th iteration.

The MATLAB, Python, and R software-based source codes of this conventional optimization algorithm for both single and multiobjective problems are made publicly available by Seyedali Mirjalili on his website at https://seyedalimirjalili.com/alo (accessed on 1 December 2021).

In [38], the authors proposed an improved version of this algorithm. In this enhanced version, the elite and selected antlions update their position using eight spiral complex paths instead of moving in randomly to improve the convergence speed and performance. These spiral trajectories include Archimedes, cycloid, epitrochoid, hypotrochoid, logarith-

mic, rose, inverse, and overshoot spirals. For an example case, the values of $R_e(t)$ and $R_a(t)$ are computed using logarithmic spiral as,

$$R_e(t) = D_1 \cdot e^{b_1 t} \cos(2\pi t_1), \ R_a(t) = D_1 \cdot e^{b_1 t} \sin(2\pi t_1), \tag{28}$$

where $D_1$, $b_1$, and $t_1$ are the parameters of logarithmic spiral (see Section 4.1.1).

Similarly, using the Archimedes spiral, the values of $R_e(t)$ and $R_a(t)$ are computed as follows:

$$R_e(t) = D_2 + b_2 \cdot t_2 \cdot \cos(2\pi t_2), \ R_a(t) = D_2 + b_2 \cdot t_2 \cdot \sin(2\pi t_2), \tag{29}$$

where $D_2$, $b_2$, and $t_2$ are the parameters of Archimedes spiral (see Section 4.1.2).

### 4.2.7. Slap Swarm Optimization Algorithm

Slap swarm optimization algorithm was developed in 2017 by Seyedali Mirjalili et al. to mimic the behavior of slap chains, which is searching for target food [89–92]. In the slap chain, the first slap is the leader, and all the other slaps follow the leader. In the algorithm, the update equations for the leader and followers' positions during the searching of target food are as follows:

$$X_i^1 = \begin{cases} F_i + r_1((UB_i - LB_i)r_2 + LB_i), & \text{if } r_3 \geq 0, \\ F_i - r_1((UB_i - LB_i)r_2 + LB_i), & \text{if } r_3 < 0, \end{cases} \tag{30}$$

$$X_i^j = 0.5(X_i^j + X_i^{j-1}), \ j \geq 2, \tag{31}$$

where $X_i^1$ and $X_i^j$ are the positions of leader and followers, $F_i$ is the target food, $LB_i$ and $UB_i$ are the lower and upper bounds of $i$th dimension, $r_1$, $r_2$, and $r_3$ are random numbers.

The MATLAB-based source code of this optimization algorithm for both single and multiobjective problems is made publicly available by the developer on his website at https://seyedalimirjalili.com/ssa (accessed on 1 December 2021). Further, the links for the source code using Python and R are also available on the same website.

However, in [65], it is stated that the conventional slap swarm optimization algorithm (SSOA) has a slower convergence and gets trapped at local optima. Thus, the authors have proposed an improved SSOA using a logarithmic spiral. In this improved algorithm, the followers' positions are updated using a logarithmic spiral as follows:

$$X_i^j = 0.5(X_i^j + X_i^{j-1}) \cdot e^{b\theta} \cdot \cos(2\pi\theta), \ j \geq 2, \tag{32}$$

where $b$ and $\theta$ are the parameters of logarithmic spiral (refer to Section 4.1.1).

### 4.2.8. Sparrow Search Optimization Algorithm

Jiankai Xue and Bo Shen proposed a sparrow search optimization algorithm in 2020 to mimic the sparrow's behaviors during group wisdom, antipredation, and foraging [93]. In this algorithm, the sparrows' population is divided into two groups of 20:80 as discovers and followers. The discover have a broad search space to search for the food and guide the followers to move towards the food source. The position update equation for the discover sparrows during the searching of target food is as follows:

$$X_{i,j}(t+1) = \begin{cases} X_{i,j}(t) \cdot \exp(-\frac{h}{\alpha \cdot M}), & \text{if } R_2 < ST, \\ X_{i,j}(t) + Q \cdot L, & \text{if } R_2 \geq ST, \end{cases} \tag{33}$$

where $X_{i,j}(t)$ and $X_{i,j}(t+1)$ are the $i$th discover sparrows' position of $j$th dimension $t$th and $(t+1)^{\text{th}}$ iterations, $h$ and $M$ are the current and maximum number of iterations, $Q$ is a uniformly distributed random number, $L$ is a row matrix with all values as one, $\alpha$ and $R_2 \in [0,1]$ are the random numbers, $ST \in [0.5, 1]$ is the safety threshold values.

The values of $R_2$ and $ST$ help indicate the safety of the food source area. Based on these values, the type of environment around the food source area, predators status, and the actions that need to be taken are classified as follows:

$$\text{Condition} = \begin{cases} \text{Safe,} & \text{No predators around, can search for food,} & \text{if } R_2 < ST, \\ \text{Unsafe, Predators around, fly to other safe area,} & \text{if } R_2 \geq ST. \end{cases} \quad (34)$$

As some of the followers closely follow the discoverers, they update their positions to move towards the discovered food source area. The position update equation for the follower sparrows towards the food source is as follows:

$$X_{i,j}(t+1) = \begin{cases} Q \cdot \exp\left(\frac{X_{worst}(t) - X_{i,j}(t)}{i^2}\right), & \text{if } i > n/2, \\ X_p(t+1) + |X_{i,j}(t) - X_p(t+1)| \cdot A^T(AA^T)^{-1} \cdot L, & \text{otherwise,} \end{cases} \quad (35)$$

where $X_{worst}(t)$ is the group's worst position at $t$th iteration, $X_p(t+1)$ is the discovers' optimal position at $(t+1)^{\text{th}}$ iteration, $A$ is row matrix of randomly assigned with 1 or $-1$. Further, $i > n/2$ indicates that the sparrows are in a danger position. Thus, the sparrows make antipredation behavior. The MATLAB-based source code for implementing this algorithm is available for registered users at https://www.mathworks.com/matlabcentral/fileexchange/88788 (accessed on 1 December 2021).

However, in [66,94], the authors proposed a variable spiral search technique for the followers to update their positions better. The position update equation of the followers using this search strategy is as follows:

$$X_{i,j}(t+1) = \begin{cases} e^{zl} \cdot \cos(2\pi l)Q \cdot \exp\left(\frac{X_{worst}(t) - X_{i,j}(t)}{i^2}\right), & \text{if } i > n/2, \\ X_p(t+1) + |X_{i,j}(t) - X_p(t+1)| \cdot A^T(AA^T)^{-1} \cdot L, & \text{otherwise,} \end{cases} \quad (36)$$

where $z$ and $l$ are the parameters of logarithmic spiral (refer to Section 4.1.1). Further, the value of $z$ is varied at every iteration, making the proposed technique a variable spiral search approach.

## 5. Application of Spiral Dynamics Optimization Algorithm

The conventional and other variants of the SDO algorithm have been applied in various fields for finding the optimal solution, as explained underneath.

### 5.1. Modeling and Controller Tuning

The application of SDO and its variants in the area of modeling and controller tuning is as follows:

- Controller tuning [95];
- Controlling robotic arm movement [96];
- Flexible manipulator system [14,16,20,26,28,34,35];
- Stair descending in a wheelchair [97,98];
- Inverted pendulum [99];
- Twin rotor systems [25,34];
- Two-wheeled robotic vehicle [39].

Hassan et al. proposed using an SDO algorithm to tune the predictive proportional-integral (PI) controller for wireless networked control systems [95]. Similarly, the authors of [96] have utilized SDO in the tuning of proportional-integral-derivative (PID) in controlling the robotic arm movement. Moreover, for both modeling and control of flexible link manipulator systems, the authors of [14] have used conventional SDO. For the same application, the authors of [20,34,35] proposed the hybridization of the SDO algorithm with BCA and BFA. The improved and adaptive version of SDO is also presented for both mod-

eling and control of a flexible link manipulator system [16,26,28]. In another application, fuzzy control of a stair descending in a wheelchair, an SDO algorithm is used for tuning of controller parameters. In [99], a hybrid algorithm using PSO and SDO is proposed for the tuning of a fuzzy controller designed for the inverted pendulum. Nasir et al. have proposed an improved SDO and hybrid algorithm using SDO and BFA to model twin rotor systems [25,34]. The hybrid SDO and BFA algorithm has also been used for controlling the two-wheeled robotic vehicles [39].

*5.2. Electrical Energy Optimization*

Similarly, the application of SDO and its variants in the area of optimizing electrical energy systems is as follows:

- Digital filters [100];
- Economic/emission dispatch [14,101,102];
- Hybrid electrical vehicles [23];
- Maximizing power production of a wind farm [103];
- Multigeneration energy system [104];
- Network with power distribution [105].

The economic and emission dispatch problems in power systems have been solved by various researchers using the SDO algorithm [14,101,102]. Similarly, an optimal strategy using the SDO algorithm is proposed for maximum power production in the wind farm [103]. A multiobjective SDO algorithm for a multigeneration energy system is presented for minimizing the total cost while maximizing energy efficiency [104]. In [105], a hybrid algorithm using SDO and BFA is proposed to optimize decentralized generation placement simultaneously. In another application, an optimal sizing strategy using the adaptive version of the SDO algorithm has been presented for hybrid electric air–ground vehicles [23]. The authors of [100] have proposed using SDO for the filter design. The algorithm achieved better performance in achieving the desired magnitude response in the multiobjective optimization task.

*5.3. Mechanical Systems Optimization*

Over the years, several mechanical systems have been optimized using the SDO algorithm. The list of applications are as follows:

- Micro-channel heat sink [29,30];
- Automation of high-rise buildings [19];
- Planar, spatial truss structures [18];
- Pressure vessel design problems [38,50];
- Welded beam design problems [50].

Cruz et al. proposed the generalized and stochastic SDO algorithms to solve microelectronic thermal management problems [29,30]. The authors of [19] have proposed a hypotrochoid SDO algorithm to optimize the sensor placement in the 632-meter-tall Shanghai Tower and compared the performance with seven optimization algorithms, including its predictors. The authors of [18] also proposed the hypotrochoid SDO algorithm for finding the optimal setting parameters of 10, 37, 52, 72, and 200-bar planar and spatial truss structures. The use of spiral equation in improving the TLBO and antlion optimization algorithms for pressure vessel design problems is presented in [38,50]. The improved TLBO algorithm using logarithmic spiral trajectory is also applied to find the optimal setting parameters for welded beam design problems [50].

*5.4. Other Optimization Problems*

The application of the SDO algorithm for other types of optimization problems are as follows:

- 2D mesh topologies [106];
- Clustering problems [33];

- Cubic polyhedral cages [107];
- Face image de-blurring [32];
- Neural network training [108,109];
- Sensor pattern sorting [110,111].

The authors of [107] are the first to showcase the problems and scope of spiral dynamics optimization applied to polyhedral cages. Another work before developing the conventional SDO algorithm is reported in [106]. Here, a heuristic spiral mapping algorithm is the first type of SDO applied for 2D mesh network topologies. For clustering problems, distributed SDO is proposed in which the population of search space is split into sub-populations [33]. Hong-Chun Jia et al. have proposed an efficient and intelligent algorithm using SDO for deep neural networks [108]. The network is to find the optimal physical health and fitness level in sports. Recently, James McCaffrey from Microsoft Research has developed the SDO algorithm in Python to train the neural network to find the optimal weights and biases values [112], the real-time implementation of a deterministic SDO algorithm using field-programmable gate arrays for spot patterns sorting in a Shack–Hartmann wavefront sensor [110].

As mentioned earlier, the SDO and its variants have been applied in various applications. The summary of all applications is given in Table 2. The table provides the details of the application system, including the dimension, software tool, cost function, type of optimization problem, and comparison techniques. In the table, SO and MO are optimization problems denoting single objective and multiobjective. The SDO validation and its variants on various benchmark functions are also detailed. It is to highlight that the most widely used error-based cost functions are: mean squared error (MSE), root mean squared error (RMSE), and the sum of squared error (SSE). Similarly, the integral error functions used in the research are integral squared error (ISE) and integral time absolute error (ITAE). The errors are computed as follows:

$$\text{MSE} \quad = \quad \frac{1}{n_s} \sum_{i=1}^{n_s} (Y_a - Y_p)^2, \tag{37}$$

$$\text{RMSE} \quad = \quad \sqrt{\frac{1}{n_s} \sum_{i=1}^{n_s} (Y_a - Y_p)^2}, \tag{38}$$

$$\text{ISE} \quad = \quad \int_{t=0}^{\infty} e(t)dt, \tag{39}$$

$$\text{ITAE} \quad = \quad \int_{t=0}^{\infty} t|e(t)|dt, \tag{40}$$

where $n_s$ is the total number of samples, $Y_a$ and $Y_p$ are the actual and predicted values, $e(t)$ is the error, the difference between actual and reference values.

**Table 2.** List of applications of SDO and its variants in various fields over the years.

| Ref. | Year | Algorithm | Validation | | Application | | | | SO/MO | Comparison |
| | | | Status | Functions | System | Dim | Tool | Cost Function | | Algorithms |
|---|---|---|---|---|---|---|---|---|---|---|
| [107] | 1997 | SDO | ✗ | - | Cubic Polyhedral Cages | - | - | - | SO | - |
| [106] | 2008 | Dynamic Spiral Mapping | ✗ | - | 2D Mesh Topologies | 3 | SMAP | Reconfiguration Time | MO | Partially, Fully DSM |
| [9] | 2010 | SDO | ✓ | Rosenbrock, Minima, Rastrigin | - | - | MATLAB | - | SO | GA, PSO, ALO |
| [11] | 2011 | SDO | ✓ | Schwefel, Minima, Rastrigin, Griewank | - | - | - | - | SO | Differential evolution (DE), PSO |
| [21] | 2012 | Adaptive SDO | ✓ | Sphere, Ackely, Grienwank | - | - | - | - | SO | SDO, Linear, Quadratic, and Exponential Adaptive SDO |
| [20] | 2012 | Hybrid SDO-BCA | ✓ | Sphere, Ackley, Rastrigin, Griewank | Flexible Manipulator System | 30 | MATLAB | ISE | SO | BFA, SDO |
| [113] | 2012 | SDO | ✓ | Sphere, Schwefel, Minima, Rastrigin, Alpine, Levy, Ackely | - | - | - | - | SO | - |
| [100] | 2013 | SDO | ✗ | - | Digital Filters | 10 | - | Weighted Magnitude, Lp Norm | SO | - |
| [14] | 2014 | Adaptive SDO, Hybrid SDO | ✓ | Rastrigin, Sphere, Griewank, Ackley | Flexible Manipulator System, Economic/Emission Dispatch, Neural Network Training | 30, 15, 9 | - | RMSE, MSE | MO | SDO |
| [114] | 2014 | Cluster-structured SDO | ✓ | Rosenbrock, Minima | - | - | - | - | SO | SDO |
| [33] | 2014 | Distributed SDO | ✗ | - | Clustering Problems | 10 | C++ | SSE | SO | SDO, Genetic K-Means |
| [34] | 2014 | Hybrid SDO-BCA | ✗ | - | Flexible Manipulator, Twin Rotor Systems | 16 | MATLAB | RMSE | MO | Recursive Least Squares (RLS), Least Mean squares (LMS), PSO, GA, Hybrid GA RLS |

**Table 2.** *Cont.*

| Ref. | Year | Algorithm | Validation | | Application | | | | SO/MO | Comparison |
|------|------|-----------|------------|-----------|------------|-----|------|---------------|-------|------------|
| | | | Status | Functions | System | Dim | Tool | Cost Function | | Algorithms |
| [24] | 2014 | Improved SDO | ✓ | Sphere, Rosenbrock, Griewank, Rastrigin | Twin Rotor System | 136 | MATLAB | Weighted RMSE | SO | BFA, SDO, Improved SDO |
| [97] | 2014 | SDO | ✗ | - | Stair Descending in a Wheelchair | 10 | MATLAB | Weighted MSE | SO | Trial and Error Method |
| [101] | 2014 | SDO | ✗ | - | Economic/Emission Dispatch | 3, 4, 60 | - | Min | SO | - |
| [115] | 2014 | SDO | ✓ | Sphere, Rosenbrock, Schwefel, Rastrigin, Ackley, Griewank, Minima, Levy, Six-hump | - | - | MATLAB | - | SO | PSO |
| [25] | 2015 | Improved SDO | ✓ | Sphere, Rosenbrock, Rastrigin, Ackley | Twin Rotor System | 50 | MATLAB | Weighted RMSE | SO | SDO |
| [96] | 2015 | SDO | ✗ | - | Controlling Robotic Arm Movement | 3 | - | Steady State Error | SO | - |
| [116] | 2015 | SDO | ✗ | - | Rectangular Microchannel Heat Sink | 5 | – | Generation Rate | MO | SA, Unified PSO |
| [26] | 2016 | Enhanced Chaotic SDO | ✓ | Sphere, Ackely, Grienwank | Single-Link Flexible Manipulator | 50 | - | MSE | SO | SDO, ABC |
| [28] | 2016 | Greedy SDO | ✓ | Sphere, Ackely, Grienwank | Single-Link Flexible Manipulator | 50 | - | MSE | SO | SDO |
| [16] | 2016 | Linear Adaptive SDO | ✓ | Sphere, Rosenbrock, Ackley, Rastrigin, Griewank, Dixon-Price, Goldstien-Price, Six-hump Camel | Flexible Manipulator Rig | 16 | MATLAB | RMSE | SO | SDO, BFA, Improved BFA |
| [117] | 2016 | SDO | ✓ | Sphere, Schwefel, Ackley, Minima, Bohachevsky, Rosenbrock | - | - | - | - | SO | ABC |

**Table 2.** *Cont.*

| Ref. | Year | Algorithm | Validation | | Application | | | | SO/MO | Comparison |
|------|------|-----------|------------|------|-------------|-----|------|---------------|-------|------------|
| | | | Status | Functions | System | Dim | Tool | Cost Function | | Algorithms |
| [110] | 2016 | SDO | ✗ | - | Shack–Hartmann Wavefront Sensor Pattern Sorting | 28 | MATLAB | RMSE | SO | B-Spline, Zernike |
| [102] | 2016 | SDO | ✗ | - | Economic/Emission Dispatch | 40 | MATLAB | Optimal Power | SO | BBO, GA, Evolutionary Algorithm (EA) |
| [105] | 2017 | Hybrid BFA-SDO | ✗ | - | Network With Power Distribution | 4 | - | Weighted Error | MO | Simulated Annealing (SA), GA, Tabu Search (TS), Hybrid BFA-PSO |
| [39] | 2017 | Hybrid SDO-BCA | ✗ | - | Two-Wheeled Robotic Vehicle Controller Tuning | 9 | - | MSE | MO | BFA, SDO |
| [95] | 2017 | SDO | ✗ | - | | 2 | MATLAB | ITAE | SO | - |
| [15] | 2017 | SDO | ✓ | Sphere, Schwefel, Minima, Levy | - | - | MATLAB | - | SO | - |
| [103] | 2017 | SDO | ✗ | - | Maximizing Power Production of Wind Farm | 50 | - | Maximum Power | SO | PSO, Game Theoretic |
| [18] | 2018 | Hypotrochoid SDO | ✗ | - | 10, 37, 52, 72, 200-bar Planar, Spatial Truss Structures | 10, 14, 8, 16, 29 | MATLAB | Min | SO | SDO, GA, PSO, DE, Hybrid SDO |
| [50] | 2018 | Spiral TLBO | ✓ | Sphere, Schwefel, Rosenbrock, Step, Quartic, Schwefel, Rastrigin, Ackley, Griewank, Penalized | Pressure Vessel, Welded Beam Design Problems | 4, 4 | MATLAB | Min | SO | TLBO, Whale Optimization Algorithm (WOA), Grey Wolf Optimizer (GWO) |
| [118] | 2018 | SDO | ✗ | - | System of Nonlinear Equations | 2, 4, 20 | C++ | Min | SO | - |
| [31] | 2018 | Stochastic SDO | ✓ | Booth, Chichinadze, Zettl, Dixon–Price, Griewank, Mishra, Wing, Rastrign | - | - | MATLAB | - | MO | Deterministic SDO, EFO, DE, Unified PSO |

**Table 2.** *Cont.*

| Ref. | Year | Algorithm | Validation | | Application | | | | SO/MO | Comparison |
| | | | Status | Functions | System | Dim | Tool | Cost Function | | Algorithms |
|------|------|-----------|--------|-----------|--------|-----|------|---------------|-------|------------|
| [37] | 2019 | Archimedean Spiral-ABC | ✓ | Griewank, Salomon, Inverted cosine, Neumaier, Beale, Colville, Kowalik, Rosenbrock, spring, Goldstein–Price | - | - | - | Mean Error | SO | Archimedean Spiral-inspired Local Search (ASLS), Modified ABC, Best-So-Far ABC |
| [42] | 2019 | Biogeography-based SDO | ✓ | Sphere, Schwefel, Axis, Quatic, Rosenbrock, Rastrigin, Griewank, Ackley, Step | CEC 2017 Benchmark Problems | - | - | Cluster count | SO | DE, BBO, Slap Swarm Optimization Algorithm (SSOA), GWO, WOA, GSA |
| [99] | 2019 | Hybrid PSO-SDO | ✗ | - | Triple-link Inverted Pendulum on Two-wheels | 4 | Simwise4D | RMSE | MO | GSA, ABC, GWO, Ant Colony Optimization (ACO), GA |
| [32] | 2019 | Iterative SDO | ✗ | - | Face Image De-blurring, Generative Adversarial Network Model | - | PyTorch | Loss Function | SO | |
| [36] | 2019 | Spiral ABC | ✓ | 10 functions of various orders | - | - | - | - | SO | ABC, Modified ABC, Best-So-Far ABC |
| [43] | 2019 | Spiral CS | ✓ | Schwefel, Quartic, Rosenbrock, Sphere, Powell, Brown, Ackley, Griewank | Spam Detection | - | Python | - | SO | PSO, DE. GA, CS, Improved CS |
| [49] | 2019 | Spiral-based SCA | ✓ | Sphere, Rosenbrock | - | - | - | - | SO | SDO, SCA |
| [19] | 2020 | Hypotrochoid SDO | ✗ | - | Automated Monitoring of High-rise Buildings | 50 | - | Modal Assurance Criterion | MO | PSO, ABC, SDO, TLBO |

Table 2. *Cont.*

| Ref. | Year | Algorithm | Validation | | Application | | | | SO/MO | Comparison Algorithms |
|------|------|-----------|------------|------------|------------|-----|------|---------------|-------|----------------------|
| | | | Status | Functions | System | Dim | Tool | Cost Function | | |
| [38] | 2020 | Improved ALO - Spiral | ✓ | 10 Unimodel, 8 Multimodel, 10 Combinatorial, 6 Multi Objective | Pressure Vessel | 4 | - | Multicriteria Function | SO, MO | Hypotrochoid, Rose, Logarithmic, Epitrochoid, Archimedes, Cycloid, and Inverse Spiral ALO |
| [104] | 2020 | Multiobjective SDO | ✗ | - | Multigeneration Energy System | - | - | Min Cost | MO | GA, PSO, EA |
| [17] | 2021 | Adaptive Hypotrochoid SDO | ✓ | Shubert, Ackley, Levy, Perm, Sphere, Trid, Booth, Beale, Powell, Shekel | - | - | - | - | SO | SDO, Adaptive SDO, Hypotrochoid SDO |
| [30] | 2021 | Reflection-based stochastic SDO | ✓ | Keane, Hosaki, Branin, Bird, Hansen, Ursem Waves, Damavandi, Giunta, Rana, 2nd Minimum | Microchannel Heat Sink | 3 | MATLAB | Minimum Entropy Generation | SO | Unified PSO, Deterministic SDO, Cuckoo search |
| [108] | 2021 | SDO | ✗ | – | Physical Fitness Determination Using Deep Neural Networks | 20 | – | RMSE | SO | GA, PSO |
| [99] | 2021 | SDO | ✗ | – | Fuzzy Control of Inverted Pendulum | 20 | MATLAB | RMSE | SO | Trial and Error, PSO |
| [35] | 2021 | Hybrid SDO-BFA | ✓ | 28 Functions | Fuzzy Control of Flexible Manipulator | 103 | MATLAB | SO | MAE | SDO, BFA, Hybrid SDO-Bacteria Chemotaxis |
| [98] | 2021 | SDO | ✗ | – | Two-wheeled Wheelchair System | 20 | MATLAB | RMSE | SO | – |
| [23] | 2022 | Adaptive SDO | ✗ | – | Hybrid Electrical Vehicles | 4 | MATLAB | Weighted Error | MO | SDO, Enhanced GA, Adaptive PSO |

## 6. Conclusions

### 6.1. Findings

SDO is a promising and fascinating algorithm that has been greatly appreciated in the literature. The SDO algorithm's advantages over other optimization algorithms lie in its simplicity, ease of implementation, the requirement of few control parameters, and better diversification and intensification strategies. This comprehensive review summarizes the research outcomes published from 1997 until January 2022. The advances and variants of SDO, including adaptive, improved, and hybrid approaches for solving various optimization problems, are critically analyzed. Further, the application of SDO and its variants in multiple fields, including modeling, controller tuning, electrical energy systems, mechanical systems, etc., is comprehensively summarized. Besides, a special interest is devoted to highlighting various nature-inspired optimization algorithms fascinated by the concept of spiral paths. This review is expected to draw the attention of the investigators, experts, and researchers to solve the optimization problems using the SDO algorithm and its variants.

### 6.2. Future Perspectives

This comprehensive review has helped open up new scopes in the field of spiral-inspired optimization and is highlighted as such underneath.

- Even though the authors have tried to avoid the issue of settling at local optima by the SDO algorithm, the issue is persisting. It requires a careful balance between exploration and exploitation phases.
- The problem of insufficient search space exploration with the conventional SDO, which uses a logarithmic spiral, can be overcome by judiciously selecting spirals. A few such spirals are Fermat, Archimedean, etc., which seem suitable in the present context to solve multiobjective problems. Specifically, the use of Fibonacci and a golden spiral is expected to solve image processing optimization problems effectively as their spiral behavior helps analyze the entire image.
- Dynamically varying control parameters in each iteration of SDO variants is still unresolved, leading to lower accuracy of the optimal solution. The selection of suitable adaptive functions for control parameters is required.
- There is a scope to improve the performance of several existing spiral-inspired optimization algorithms either by utilizing the spiral position update equation of SDO or using other spiral trajectories. Further, the natural behavior of nonspiral-inspired algorithms can be modified using spiral paths for better accuracy in the optimal solution.
- The lack of a mathematical model for complex spiral trajectories, such as the Celtic spiral, limits its use for better search space exploration. Hence, the development of suitable models for such a complex spiral trajectory is expected to enhance the SDO algorithm's exploration performance.

**Author Contributions:** Conceptualization, K.B.; proofreading, guidance, and regular feedback, B.R.P., M.B.O. and R.I.; writing—original draft preparation, M.B.O. and K.B.; writing—review and editing, B.R.P.; supervision, R.I.; project administration and funding acquisition, M.B.O. and R.I. All authors have read and agreed to the published version of the manuscript.

**Funding:** This research was funded by Yayasan Universiti Teknologi PETRONAS Fundamental Research Grant (YUTP-FRG) number 015LC0-362.

**Institutional Review Board Statement:** Not applicable.

**Informed Consent Statement:** Not applicable.

**Data Availability Statement:** No new data were created or analyzed in this study. Data sharing is not applicable to this article.

**Conflicts of Interest:** The authors declare no conflict of interest.

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
