# Peer review of "Recent Advances and Applications of Spiral Dynamics Optimization Algorithm: A Review"

_fractalfract, doi:10.3390/fractalfract6010027_

Round 1

Reviewer 1 Report

This paper reviews the recent advances of the SDO algorithm, the SDO algorithm, and its application in various areas. Generally, the manuscript is well organized. It presents a comprehensive reference to many related studies. The characteristic of the SDO and its recent variants as well as applications are well categorized and listed. However, it could be better if each of these variants and applications are described briefly instead of just listing the related works.

Besides, since readers would pay great attention to the Future Perspectives (subsection 6.2), it should be discussed in more detail (the future trends should come with some real-world problems or imperative needs).

Reviewer 2 Report

This paper reviews the development, application, and future perspectives of spiral dynamics optimization algorithms. The author has investigated a large number of literature, which is meaningful for other researchers in this field. There are some suggestions: 

  1. Fractal Fract mainly focuses on fractals, fractional calculus, and their applications in different fields. The relationship between spiral dynamic optimization and fractal theory should be emphasized in this paper.
  2. Section 4 talks about “Application of Spiral Dynamics Optimization Algorithm”, but section 5 talks about “Spiral Path Inspired Optimization Algorithms”. I think this order is inappropriate. Maybe the application of the algorithm should be placed at the end of the article.
  3. In the introduction, the advantages of the physics-based algorithm should be specifically illustrated compared with other algorithms, such as swarm intelligence-based algorithms.
  4. In section 5, this part presents the formula of algorithms and the code but there is less content describing the disadvantages of these algorithms. The drawbacks of spiral path-based optimization algorithms should be discussed.
  5. In section 2, in line 119, “the hypotrochoid can search most of the area in the search space”, but it is not clear in Figure 4 that this algorithm can search most of the area. Maybe a clearer figure in which the red line explores more space is better.
  6. In Figure 6, the words should be revised, such as “using”, “uses”. Please be consistent.
